# Generative Adversarial Networks for Coronary CT Angiography Acquisition Protocol Correction with Explicit Attenuation Constraints

**Rudolf L. M. van Herten**[1,2]                                       R.L.M.VANHERTEN@AMSTERDAMUMC.NL
**Louis D. van Harten**[1,2]                                          L.D.VANHARTEN@AMSTERDAMUMC.NL
**R. Nils Planken**[3]                                                R.N.PLANKEN@AMSTERDAMUMC.NL
**Ivana Išgum**[1,2,3]                                                I.ISGUM@AMSTERDAMUMC.NL

[1] *Department of Biomedical Engineering and Physics, Amsterdam University Medical Center - location University of Amsterdam, the Netherlands.*

[2] *Informatics Institute, University of Amsterdam, the Netherlands.*

[3] *Department of Radiology and Nuclear Medicine, Amsterdam University Medical Center - location University of Amsterdam, The Netherlands.*

**Editors:** Accepted for publication at MIDL 2023

## Abstract

The image quality of coronary CT angiography (CCTA) is important for the correct diagnosis of patients with suspected coronary artery disease, which is heavily influenced by image acquisition. Timing of the contrast media injection specifically influences the level of arterial enhancement, and it is aimed to allow optimal assessment of the coronary artery morphology. However, a consensus on an optimal acquisition protocol that can account for the large variety in patient cohorts has not been reached, commonly resulting in suboptimal arterial enhancement. In this work, we propose a generative adversarial network for the retrospective correction of contrast media attenuation in CCTA, thus reducing the dependency on an optimal timing protocol at acquisition. We develop and evaluate the method in a set of 1,179 CCTA scans with varying levels of contrast enhancement. We evaluate the consistency of intensity values in the coronary arteries and evaluate performance of coronary centerline extraction as a commonly performed analysis task. Results show that correction of contrast media attenuation values in CCTA scans is feasible, and that it improves the performance of automatic centerline extraction. The method may allow improved analysis of coronary arteries in CCTA scans with suboptimal contrast enhancement.

**Keywords:** Contrast media timing, coronary attenuation, coronary CT angiography, deep learning, generative adversarial networks

## 1. Introduction

Coronary CT angiography (CCTA) is an effective modality for the non-invasive diagnosis of (suspected) coronary artery disease (Hoffmann et al., 2012; Mortensen et al., 2020). Through contrast enhancement it visualizes the coronary artery lumen, and allows for the evaluation of arterial stenosis and atherosclerotic plaque. These are important for patient management (Cury et al., 2016). However, clinical assessment of CCTA scans heavily relies on scan quality, which is affected by a number of factors, including timing of contrast media injection, cardiac motion, and blooming artifacts (Dey et al., 2008). Insufficient arterial enhancement ($\leq 300$ Hounsfield Units (HU)) typically leads to unreliable and often

overestimated evaluation of the stenosis degree (Cademartiri et al., 2006, 2008). Conversely, elevated levels of coronary enhancement (i.e. $\geq 500$ HU) may cause beam hardening and partial volume effects, resulting in an underestimation of stenosis degree (Horiguchi et al., 2007; Fei et al., 2008). Ideally, for the analysis of coronary arteries a contrast medium injection protocol should always result in coronary artery attenuation values approaching 400 HU. Consistency in contrast enhancement of the coronary arteries in CCTA scans depends on the chosen image acquisition protocol. Several contrast media injection protocols optimizing coronary enhancement were proposed (Fleischmann, 2010; Kawaguchi et al., 2014; Sandfort et al., 2020), but they may be limited in accounting for differences in patient cohort characteristics.

As an alternative to the optimization during image acquisition, recent advances in deep learning have allowed for the retrospective adjustment of previously acquired medical images. Generative modelling has been explored to address image quality in various images and modalities (Ghekiere et al., 2017; Wolterink et al., 2021). Specifically, in CCTA scans, generative models have been used to reduce image noise (Gondara, 2016; Kang et al., 2019), reduce motion artifacts (Jung et al., 2020), and perform super-resolution (Sun and Ng, 2022). However, the correction of the contrast media enhancement levels in CCTA has seen little attention thus far.

Hence, we propose a method for the retrospective adjustment of coronary artery attenuation values in CCTA. We employ a generative adversarial network (GAN) which, given a CCTA scan with suboptimal contract enhancement, synthesizes a CCTA scan with contrast intensity levels in an optimal range (Goodfellow et al., 2020). We develop and evaluate the method with CCTA scans of 1,179 patients suspected of coronary artery disease. To evaluate the performance of the method, we compare the consistency of the intensity values in the main coronary arteries before and after arterial contrast correction. Moreover, we compare performance of coronary artery centerline extraction, a commonly performed downstream task in scans before and after correction.

## 2. Data

### 2.1. Patient and Image Data

This study retrospectively included 1,179 CCTA scans of patients with suspected coronary artery disease from clinical routine in the Amsterdam University Medical Center - Location UvA. The need for informed consent was waived by the Institutional Ethical Review Board. CCTA exams were acquired on a Siemens Somaton Force CT scanner (Siemens Healthineers, Erlangen, Germany). The tube voltage ranged from 70 to 120 kVp and the tube current ranged from 296 to 644 mAs. Scans were ECG triggered and have an in-plane resolution of 0.26-0.46 mm$^2$ and a slice thickness and increment of 0.6 mm. During acquisition, contrast medium was injected for optimal enhancement of the coronary arteries.

For all patients, coronary artery ostia and centerlines were extracted using the CNN-based orientation classifier described by (Wolterink et al., 2019b). The mean and standard deviation of the attenuation at the aortic root were determined by extracting and evaluating 3D patches of size 9.5 mm $\times$ 9.5 mm $\times$ 9.5 mm centered around the ostia. This was done by identifying the Gaussian distribution describing the contrast agent in the patch histogram (see Appendix 7.1). Based on the mean contrast attenuation $\mu_{HU}$, scans were subsequently

divided into a group of *optimal* CCTA scans ($300 < \mu_{HU} < 500$) and a group of *suboptimal* CCTA scans ($\mu_{HU} \leq 300$, $\mu_{HU} \geq 500$). This resulted in a total of 402 optimal and 777 suboptimal scans.

A total of 96 patients with suboptimal levels of contrast enhancement were prospectively set apart for evaluation. For this set, automatic coronary artery lumen and myocardium segmentations were obtained through the respective methods described by (Wolterink et al., 2019a) and (Bruns et al., 2022).

## 2.2. Reference Standard

To evaluate the completeness of automatic coronary artery centerline tracking, a trained observer manually placed markers in the three major coronary arteries (i.e. the right coronary artery (RCA), the left anterior descending (LAD), and the left circumflex (LCx)) in CCTA scans from the hold-out test set comprising 96 scans (see Section 2.1). These markers were placed along the centerline of the coronary lumen for each artery starting from the ostia, resulting in a total of 1,148 markers. Similarly, a total of 785 markers were placed in 68 scans derived from the optimal CCTA set.

## 3. Method

We propose a GAN for the automatic adjustment of contrast in suboptimally acquired CCTA scans. GANs typically consist of two main parts: a generator network and a discriminator network. For our purposes, the generator generates images with optimal contrast from images with suboptimal contrast, while the discriminator tries to distinguish generated images from real optimal images. Once trained, the generator accepts axial CCTA slices as an input and returns a synthesized version of the slice with a contrast level within the optimal range of approximately 400 HU.

Inspired by previous work for coronary artery calcium (CAC) scoring that features decomposition of non-contrast CT with CAC into an image without CAC and a corresponding CAC map (Van Velzen et al., 2022), we assume that suboptimally acquired CCTA scans ($I_{sub}$) can be decomposed into a CCTA scan with optimal contrast ($I_{opt}$) and a superimposed contrast offset mask ($M_{off}$), i.e. $I_{sub} = I_{opt} + M_{off}$. As such, the task of the generator is simplified to only compute the difference between the two CT scans. The proposed decomposition is illustrated in Figure 1.

The generator architecture used to produce $M_{off}$ employs a CNN backbone consisting of 6 ResNet-blocks, which has been shown to be capable of performing high-quality image-to-image translation (He et al., 2016; Zhu et al., 2017). For the discriminator network we employ a PatchGAN, which learns to classify whether $70 \times 70$ overlapping patches are real ($I_{opt}$) or fake ($\hat{I}_{opt}$) images with an optimal contrast level (Isola et al., 2017). The fully convolutional nature of PatchGAN allows it to operate on arbitrary image sizes while requiring fewer parameters than a full-image discriminator. Since the discriminator produces an output matrix for the overlapping patches, image labels are projected to match the output size of the discriminator network before computation of the loss.

The loss function employed in this work comprises several terms for the generation of realistic optimally acquired CCTA scans, and is defined as:

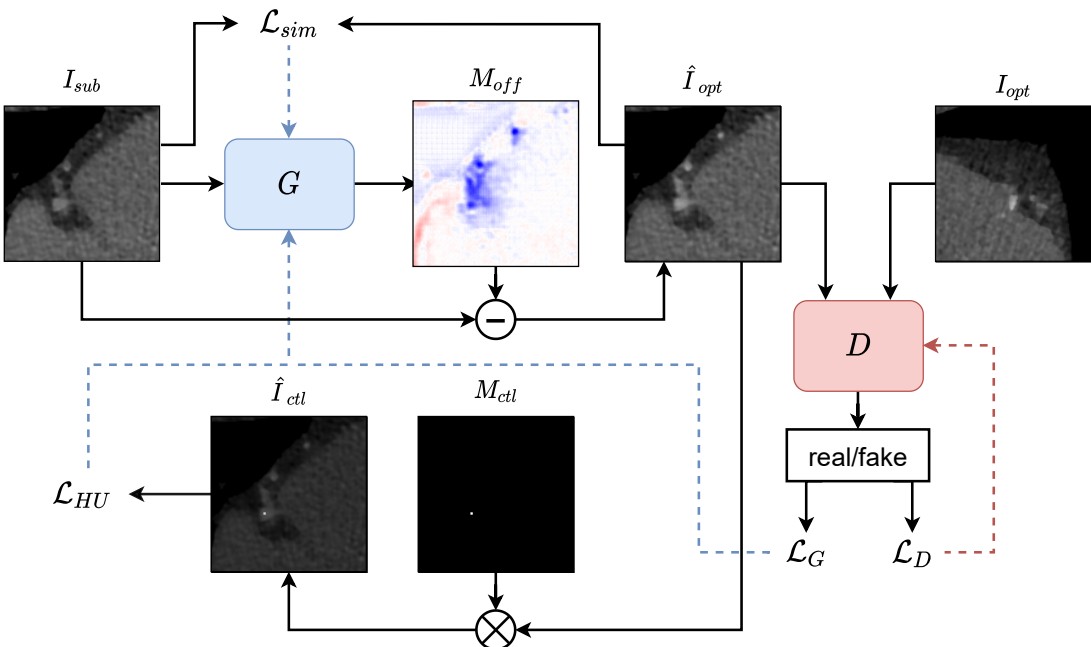

Figure 1: Overview of the proposed methodology. A generative model $G$ analyzes CCTA images acquired with suboptimal contrast media timing ($I_{sub}$), and predicts a contrast offset mask ($M_{off}$) that is subtracted from the input image to obtain a synthetic optimally acquired image ($\hat{I}_{opt}$). Optimal contrast is enforced by constraining contrast values in a coronary artery centerline mask ($M_{ctl}$) to operate within a certain range of HU values ($\mathcal{L}_{HU}$). Realistic images are encouraged by training a discriminator $D$ to distinguish real optimally acquired CCTA images ($I_{opt}$) from generated ones with $\mathcal{L}_G$ and $\mathcal{L}_D$, while similarity between generator input and output is encouraged through $\mathcal{L}_{sim}$.

$$\mathcal{L} = \mathcal{L}_G + \mathcal{L}_D + \mathcal{L}_{sim} + \mathcal{L}_{HU} \tag{1}$$

Here, $\mathcal{L}_G$ and $\mathcal{L}_D$ describe the adversarial loss used in GAN optimization. The similarity loss $\mathcal{L}_{sim}$ directly minimizes differences between $I_{sub}$ and $\hat{I}_{opt}$, and is defined as the negative normalized cross-correlation between the two images (De Vos et al., 2019). Hence,

$$\mathcal{L}_G, \mathcal{L}_D = \min_G \max_D \mathbb{E}_{I_{opt} \sim p_{\text{data}}(opt)}[\log D(I_{opt})] + \mathbb{E}_{I_{sub} \sim p_{\text{data}}(sub)}[1 - \log D(G(I_{sub}))] \tag{2}$$

$$\mathcal{L}_{sim} = -NCC(I_{sub}, \hat{I}_{opt}) \tag{3}$$

The combination of these losses ensures the generation of images that feature the high-frequency details derived from the input scan, yet appear similar to optimally acquired CCTA scans.

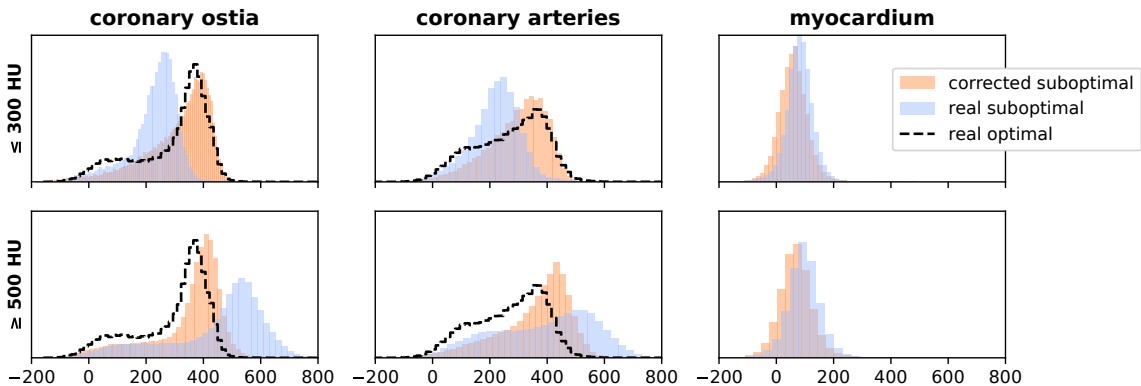

Figure 2: Attenuation distributions in the coronary ostia, coronary arteries, and the myocardium, before and after applying the method. The top row describes the test set images for which contrast level was below the optimum range ($\leq 300$ HU) in the original scan, while the bottom row describes images which had a contrast level above the optimum range ($\geq 500$ HU).

Since we are interested in enforcing a specific range of contrast attenuation in the coronary arteries and by extension the aortic root, the final constituent term of the loss function $\mathcal{L}_{HU}$ explicitly penalizes outliers in the optimal contrast range along coronary artery centerlines. This is achieved by providing a mask for each input CCTA slice in which centerline point locations are marked. The mask of $2 \times 2$ voxels is placed around the identified centerline point. By applying this mask to $\hat{I}_{opt}$, predicted attenuation values along coronary artery centerlines are extracted ($\hat{I}_{ctl}$) and subsequently penalized when contrast values are not within an optimal range. This results in the following loss component:

$$\mathcal{L}_{HU} = \sum_{j=1}^{N}(\min(\hat{I}_{ctl}^{j}, HU_{min}) - HU_{min})^2 + \sum_{j=1}^{N}(\max(\hat{I}_{ctl}^{j}, HU_{max}) - HU_{max})^2 \quad (4)$$

where $N$ is the number of masked centerline voxels in a training batch, and $HU_{min}$ and $HU_{max}$ describe a lower and upper bound for the expected contrast range. The implementation of the presented method is publicly available[1].

## 4. Evaluation

The proposed method is evaluated qualitatively by observing the shifts in frequency distribution of attenuation values in the CT image. This is performed for three cardiac structures: the ostia, the coronary artery lumen, and the myocardium. We further evaluate the method quantitatively by assessing the effect of contrast enhancement correction on automatic coronary artery centerline tracking. Centerline tracking typically fails when contrast levels are

---

1. https://github.com/RoelvH97/ContrastGAN

outside of an expected range, and should therefore benefit from contrast correction. We perform centerline tracking in the original and in the corrected images and evaluate the completeness of the extracted coronary artery trees. The marker recovery rate along the major coronary arteries is evaluated before and after GAN correction. For the marker recovery rate, a hit is defined as a manually identified marker being within a 5 mm radius of an automatically extracted centerline point. Finally, we compare attenuation values at marker locations before and after the correction in terms of histogram intersection.

## 5. Experiments and Results

### 5.1. Experimental Setting

The 1,083 images in the training set were divided into two subsets: 80% for training and 20% for validation. Performance on the correction of suboptimal contrast levels in the validation set was tracked during training, allowing for hyperparameter tuning and the selection of the best-performing generative model.

The generator network was trained on minibatches, which were balanced to contain axial patches cropped around coronary artery centerlines and patches cropped around random locations in the image. Batches were further balanced to contain both patches below and above the optimal range of arterial enhancement. Training patches were of size $128 \times 128$ voxels, and patches cropped around centerlines were randomly rotated or flipped with $p = 0.5$. Batches for training the discriminator were sampled similarly, but were balanced with respect to suboptimal and optimal acquisition. Image values were divided by a factor of 600 to normalize GAN inputs. In tandem with the tanh activation function of the generator, this allows for a maximum contrast offset of 600 HU.

During training, a batch size of 64 was employed for both the generator and the discriminator. Both networks were optimized using the Adam optimizer with a learning rate of 2e-4 and parameters $\beta_1 = 0.5$ and $\beta_2 = 0.999$ (Kingma and Ba, 2014). Networks were trained for a total of 100,000 iterations, with the learning rate being multiplied with a factor of $\gamma = 0.1$ at iterations [60,000; 80,000]. We further defined $HU_{min}$ and $HU_{max}$ to be 350 HU and 450 HU respectively to ensure adequate attenuation along coronary arteries. At test time, the generator processes full axial CCTA input slices of size $512 \times 512$ voxels in a single forward pass. All methods and optimizations were implemented with the PyTorch deep learning library (Paszke et al., 2019). The proposed method is compared to a histogram matching baseline, for which the details are listed in Appendix 7.3.

### 5.2. Results

Attenuation frequency distribution shifts in three cardiac structures are shown by the histogram plots in Figure 2, displaying distributions before and after applying the method. For both the coronary ostia and arteries, a shift of intensity values resulting from the correction is observed towards the contrast distribution in optimally acquired scans. According to expectation, a shift is not observed in the myocardium, which is largely unaffected by contrast enhancement. Two examples of CCTA slice correction are presented in Figure 3.

Figure 4 displays the impact of contrast correction on automatic coronary artery centerline extraction. The overall marker recovery rate of manually identified markers in the test

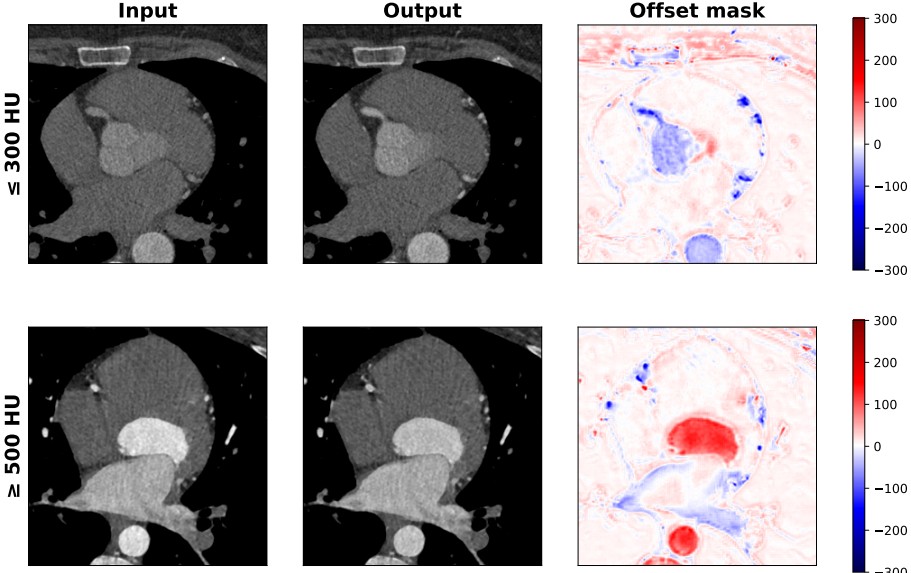

Figure 3: Outputs of the proposed method for an image with contrast level below (top) and above (bottom) the optimal range. Note that the highlighted regions in the offset mask strongly affect image contrast in the aortic root and coronary arteries, with blue denoting an increase in attenuation and red denoting a decrease.

set was 88% before correction, 77% after histogram matching and 90% after GAN correction. For both the LAD and the LCx, an increase in marker recovery rate is also observed after GAN correction, while a slight decrease in performance is observed for the RCA. Histogram intersection of markers annotated in suboptimally acquired images with markers annotated in images with optimal contrast enhancement was 0.49 before correction, 0.57 after histogram matching, and 0.73 after GAN correction. Average and standard deviation of the attenuation values at marker locations were $423 \pm 182$ HU before and $405 \pm 101$ HU after GAN correction. An extensive quantitative evaluation is presented in Appendix 7.4.

## 6. Discussion and Conclusion

This work proposed a method for correcting contrast media enhancement in CCTA scans. The method leverages a GAN to learn a difference map between suboptimal and optimal levels of contrast enhancement, and is constrained to force attenuation values along coronary artery centerlines within a specific range. This allows for the adjustment of previously acquired CCTA scans, thus rendering a realistic image with optimal contrast attenuation.

The qualitative evaluation reveals that the method learns to accurately target volumes affected by contrast enhancement, and does not influence the attenuation level in tissue which is not directly impacted by angiography (Figure 2). Furthermore, the results indicate that attenuation value distribution shifts in the coronary ostia and coronary arteries result

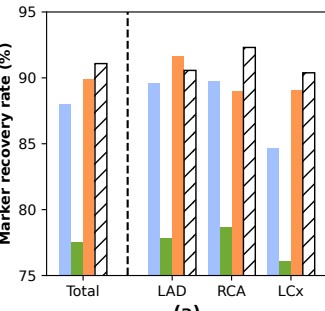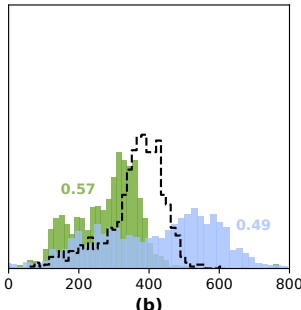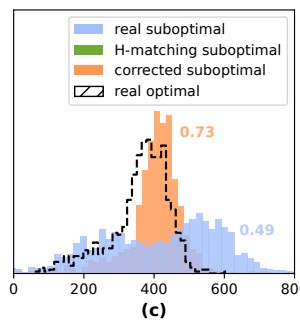

Figure 4: Results for coronary artery centerline extraction. (a) Marker recovery rate of automatic coronary artery centerline extraction for manually placed markers along the three major coronary arteries in the test set. Bars indicate performance on suboptimal scans (blue), after applying histogram matching (green), after applying the proposed method (orange), and performance on optimal scans (hatched). (b) Attenuation values at marker locations in HU, before (blue) and after (green) histogram matching. (c) Attenuation values after applying the proposed method (orange). Values in (b) and (c) indicate histogram intersection with markers annotated in optimal images (hatched).

in a distribution which closely resembles real CCTA scans with an optimal level of contrast enhancement. Though the myocardium is eventually affected by contrast enhancement as well, training did not differentiate between early and late enhancement scans, which may be an interesting topic for future research.

Results on the extraction of coronary artery centerlines indicate a better overall recall for contrast-corrected scans. The largest improvement is observed in the LCx, which may be attributed to the fact that only a small percentage of the population has a left-dominant coronary circulation. As such, the LCx is typically smaller and more difficult to distinguish, therefore benefiting more from contrast correction. The RCA on the other hand is typically subject to motion artifacts, which our method does not correct for. This may explain why no improvement is found for the recall of RCA markers.

A limitation of the proposed method may be that it only processes 2D axial slices. Consequently, the network is only able to learn the appearance of arterial structures in a 2D plane, therefore missing potentially valuable 3D information on the connectivity of coronary arteries. Furthermore, a larger field of view may help the model to identify other cardiac structures relevant for attenuation correction. Nonetheless, results indicate that explicit penalization of values along coronary artery centerlines helps the model in finding an optimal distribution of contrast enhancement that ignores most 2D noise. Future work could investigate whether a 3D approach may help the method in creating even better corrections for contrast enhancement.

Given that the method employs a GAN, it allows for training with unpaired data. Obtaining paired data for training would be impossible because voxel level spatial alignment of

a patient is infeasible. As such, it offers advantages over standard CNN architectures requiring end-to-end data. Future work might investigate whether advanced synthesis methods may contribute to performance improvement (Ho et al., 2020; Rombach et al., 2022).

In conclusion, this study presented a method for the retrospective correction of CCTA scans with suboptimal levels of coronary artery contrast enhancement by generating contrast offset masks using a GAN. Contrast enhancement correction is shown to be feasible and produce realistic images within an optimal range of attenuation values. The method may extend applicability of automatic coronary artery analysis to CCTA scans with suboptimal contrast enhancement.

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

## 7. Appendix

### 7.1. Pre-processing

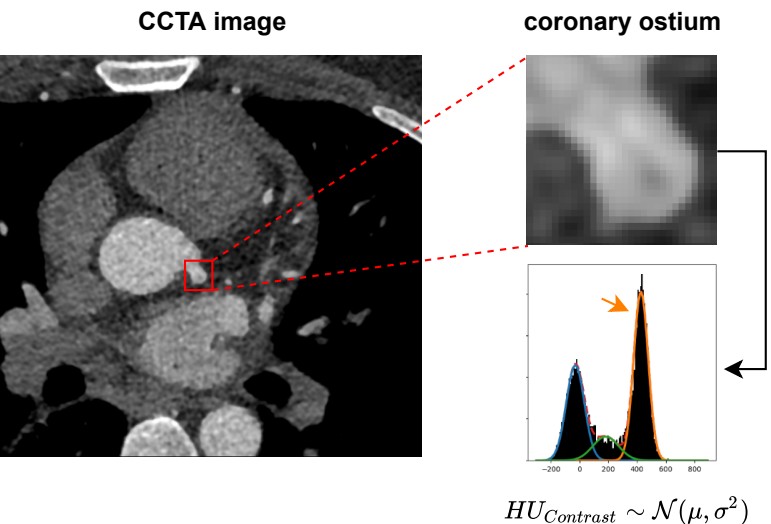

$$HU_{Contrast} \sim \mathcal{N}(\mu, \sigma^2)$$

Figure 5: Suboptimal contrast detection pipeline. Coronary ostia are first extracted from the CCTA through an automated method (Wolterink et al., 2019b). A 3D patch is subsequently extracted around both ostia, for which the intensity histogram is computed (right). By representing the histogram as a mixture of Gaussians, the histogram peak describing the contrast media can be identified. The value of this peak is used to label CCTA scan as either suboptimal or optimal.

## 7.2. Evaluation

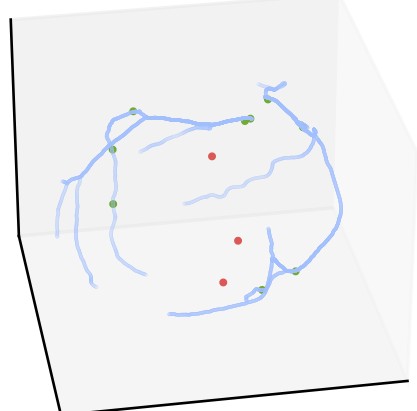

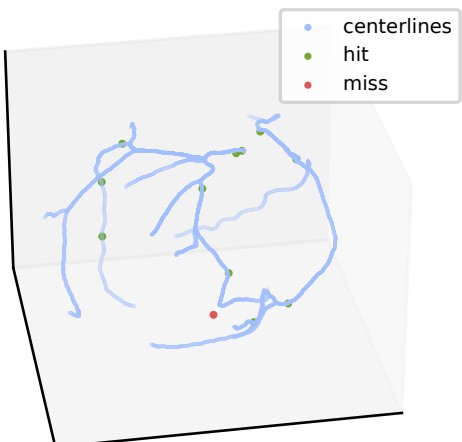

Figure 6: Evaluation of the marker recovery rate. Blue lines indicate automatically extracted centerlines, while green and red dots indicate manually placed recovered and missed markers respectively. In this example, automatic centerline tracking of the LCx fails in the original scan (left) as indicated by the red dots. Upon correction (right), a large part of the LCx is recovered, resulting in a higher marker recovery rate. Note that the missed marker on the corrected scan (right) is located in an undetected coronary branch.

## 7.3. Histogram Matching

For histogram matching, a template histogram is defined based on a selection of 10 scans from the optimal training images. Mean contrast at the coronary ostia for the selected images was in a range of $350 - 450$ HU with a maximum standard deviation of 25 HU. All histograms of test set images were matched to this template histogram, for which distribution shifts are displayed in Figure 7. Qualitative examples of histogram matching are displayed in Figure 8.

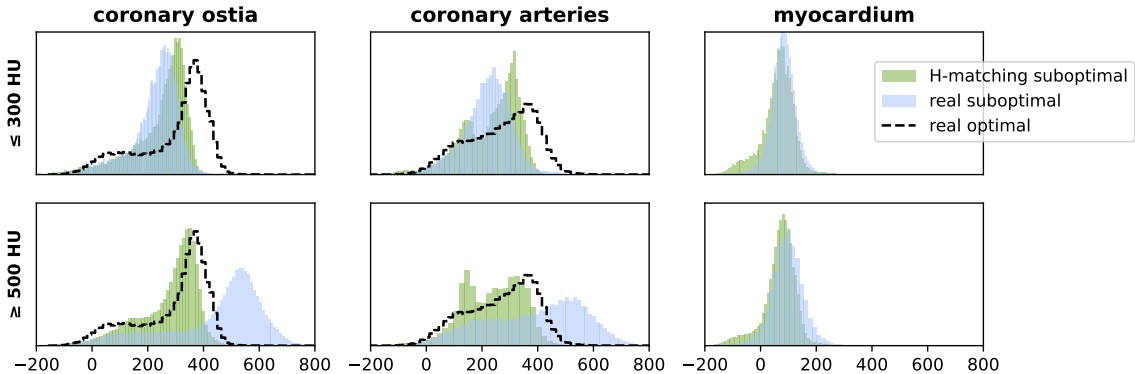

Figure 7: Attenuation distributions in the coronary ostia, coronary arteries, and the my-ocardium, before and after applying histogram matching. The top row describes the test set images for which contrast level was below the optimum range ($\leq 300$ HU) in the original scan, while the bottom row describes images which had a contrast level above the optimum range ($\geq 500$ HU).

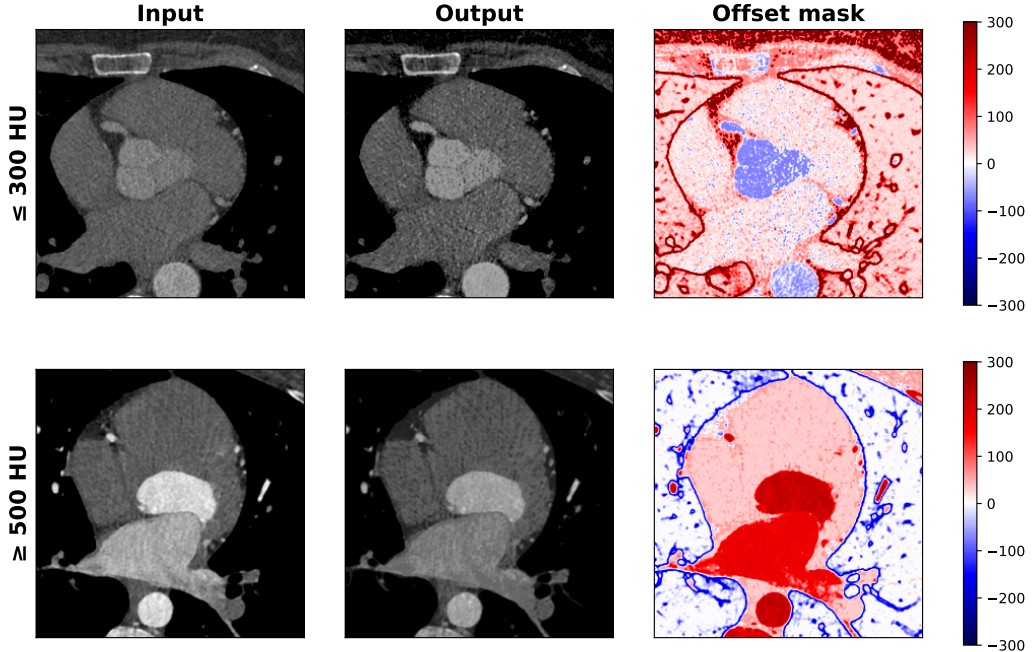

Figure 8: Output examples for histogram matching. Note that the global intensity mapping enhances the existing noise in the input images, thus typically deteriorating image quality.

## 7.4. Extended results

Table 1 presents the full quantitative results comparing histogram matching to the proposed method. The results show that the proposed method outperforms histogram matching in most cases, achieving the highest overall histogram intersection values and marker recovery rates. Histogram matching is shown to have a higher histogram intersection in the range of $\geq 500$ HU for the ostia and arteries, which is likely due to an easier separability of arterial values. Despite this, the marker recovery rate for histogram matching is lowest across all major coronary arteries and methods. Hence, the noise introduced by global intensity mapping in histogram matching is shown to negatively impact the performance of downstream tasks, despite pushing the overall attenuation distribution to a more optimal range.

Table 1: Quantitative results divided by suboptimal HU intensity value range. Histogram intersection results are listed for the coronary ostia, coronary arteries, and manually placed markers. It is defined as the intersection with the histogram of the same structures in the set of optimally acquired images. Marker recovery rate is further presented separately for the three major coronary arteries.

| HU range | Method | Histogram intersection | | | Marker recovery (%) | | |
|---|---|---|---|---|---|---|---|
| | | Ostia | Arteries | Markers | LAD | RCA | LCx |
| $\leq 300$ | - | 0.47 | 0.65 | 0.37 | 79.1 | **92.2** | 72.0 |
| | Histogram matching | 0.56 | 0.73 | 0.48 | 70.1 | 85.9 | 67.6 |
| | Proposed (GAN) | **0.86** | **0.87** | **0.75** | **83.6** | 89.1 | **82.4** |
| $\geq 500$ | - | 0.39 | 0.59 | 0.46 | 91.8 | **89.2** | 87.3 |
| | Histogram matching | **0.76** | **0.81** | 0.59 | 79.4 | 77.2 | 77.8 |
| | Proposed (GAN) | 0.73 | 0.67 | **0.67** | **93.4** | 88.9 | **90.5** |

