# OpenReview forum: "Generative Adversarial Networks for Coronary CT Angiography Acquisition Protocol Correction with Explicit Attenuation Constraints"
_MIDL.io/2023/Conference — MIDL 2023 Poster_

### Official Review · Reviewer_GPE7 · 2023-01-31

**Confidence:** 3
**Preliminary Rating:** 4
**Recommendation:** Poster

**Summary:**

The paper proposes to correct the contrast of coronary CT angiography scans to improve the downstream task of automatic centerline extraction. To do so, a Generative Adversarial Network is trained to predict a "contrast offset mask" to images with suboptimal contrasts, and to discriminate between contrast-corrected and real optimally-contrasted images.

 ~1200 scans were acquired for the task, which include ~400 optimally-contrasted scans and ~800 sub-optimally contrasted scans.

To evaluate the performance of the method, a qualitative assessment of corrected contrasts versus uncorrected constrats was performed, and a quantitative assessment of centerline extraction on corrected versus uncorrected scans was done.

**Strengths:**

The paper is well written and its methodological developments are sound. The use of a GAN to perform contrast correction is interesting and removes the need for paired images, or having to artificially degrade the contrast of scans. The method is well described and the experimental parameters are clearly communicated. The method uses a large fairly dataset and makes use of it for hyper-parameter search.

**Weaknesses:**

The main weakness of the proposed method comes from the lack of robust validation of results. Most importantly, the method should have been compared to classical approaches of contrast improvements or DL-based methods. As it stands now, while the method seem to provide improvements, we have no way of knowing if the improvements are even better than naive methods.

Second, the reported "metrics" are lacking. The qualitative assessment could easily become quantitative by computing one of many histogram similarity metric. The quantitative assessment is lacking as only recall is reported, without precision.

**Deanonymize Review:**

no

**Paper Type:**

methodological development

**Questions To Address In The Rebuttal:**

- What other DL-based methods could be used for contrast correction ?
- What other "classical" image-processing methods could be used for contrast correction ?
- Why were they not used for comparison ?
- What can be done to improve the robustness of the validation of your results ?

---

### Official Review · Reviewer_r1cm · 2023-02-02

**Confidence:** 4
**Preliminary Rating:** 3
**Recommendation:** Poster

**Summary:**

The paper presents a GAN approach to adjust image contrast on coronary CT angiography. The authors trained on a set of ~1000 image of optimal and non-optimal contrast (not paired) to identify a mapping procedure to achieve optimal contrast. Efficacy was assess via comparison to manual landmarks on 96 scans with a total of 1148 manual landmarks via recall. Recall improved for 2 of 3 structures.

**Strengths:**

+ The writing is clear.
+ The problem domain is important.
+ A reasonable dataset exists.
+ The application domain appears appropriate for the problem.
+ Problem definition figures are very clear.

**Weaknesses:**

- No qualitative imaging results are shown.
- Recall is an unusual metric to use in isolation.
- No 3D center lines are shown.
- No magnitude of effect changes on the extended centerline are shown.
- No recall is shown on the optimal contrast approaches (baseline optimal error rates).


**Deanonymize Review:**

no

**Detailed Comments:**

The rebuttal clearly addressed concerns raised.

**Paper Type:**

validation/application paper

**Questions To Address In The Rebuttal:**

What is the innovation of the network design? Is there a baseline approach?

How effective is this method? What is the performance of the landmarks on the optimal dataset?

How much does this method change the overall path of the desired paths? A qualitative analysis? A secondary interpretation function?

What is the qualitative change in image appearance?

---

### Official Review · Reviewer_EMPP · 2023-02-02

**Confidence:** 4
**Preliminary Rating:** 4
**Recommendation:** Poster

**Summary:**

This work proposes to use a GAN to retrospectively correct contrast media attenuation in coronary CT angiography, reducing the dependency on optimal timing protocol and improving performance of coronary centerline extraction. The evaluation was conducted on a large dataset through a qualitative approach. The results show the potential of the proposed method for clinical use.

**Strengths:**

- This study employs deep learning for an innovative application of contrast media attenuation correction for CCTA.
- The methodology and evaluation techniques are solid.
- The results are encouraging and show the potential of the proposed method being useful in clinical settings.
- The manuscript is well-written and easy to follow.

**Weaknesses:**

- Section 5.1 shows that the training was done with patches, not whole images. However, it is not evident how the inference was conducted. The generator can cause variations in intensity distribution, so simply combining the patches may lead to unwanted results. If the inference was still done with patches, the authors must explain how the patches were combined, for example by using a sliding-patch inference approach or through stitching or averaging.
- The absence of baseline methods is a shortcoming. It would be beneficial if the proposed method could be compared with the traditional, non-DL methods discussed in the introduction section.
- The evaluation method used is somewhat subjective. The authors stated that a "trained" observer performed the evaluation qualitatively. However, it is uncertain whether this observer has the proper qualifications to evaluate a medical imaging technique, as it is not standard practice for non-medical professionals to do so.

**Deanonymize Review:**

no

**Paper Type:**

validation/application paper

**Questions To Address In The Rebuttal:**

- Please explain how inference was done.
- It would strengthen the paper to include several baseline methods.
- Please specify the qualifications of the "trained" observer. If this observer is not a medical expert, it is recommended to have the evaluation conducted by multiple observers and present results from multiple evaluations, as the bias of a single observer may impact the results.

---

### Meta-Review · Area_Chair_Wgp5 · 2023-02-24

**Recommendation:** Accept (Poster)
**Confidence:** 4

**Metareview:**

The authors propose GANs for contrast media attenuation correction in CCTA.
The major concern raised by the reviewers was absence of a baseline method. The authors noted the lack of research for automated contrast media attenuation correction in CCTA, and compared it against histogram equalization. There were other minor concerns that were also addressed by the reviewers.
The paper is easy to follow, and since the authors addressed all major concerns raised by the reviewers. Therefore, I recommend acceptance.